# A Noval Weighted Meta Graph Method for Classification in Heterogeneous Information Networks

**Jinli Zhang** [1,2] iD, **Tong Li** [1,*] iD, **Zongli Jiang** [1] iD, **Xiaohua Hu** [2] iD and **Ali Jazayeri** [2] iD

[1] Faculty of Information Technology, Beijing University of Technology, Beijing 100124, China; zhangjl86@emails.bjut.edu.cn (J.Z.); jiangzl@bjut.edu.cn (Z.J.)

[2] College of Computing and Informatics, Drexel University, Philadelphia, PA 19104, USA; xh29@drexel.edu (X.H.); aj629@drexel.edu (A.J.)

[*] Correspondence: litong@bjut.edu.cn

**Abstract:** There has been increasing interest in the analysis and mining of Heterogeneous Information Networks (HINs) and the classification of their components in recent years. However, there are multiple challenges associated with distinguishing different types of objects in HINs in real-world applications. In this paper, a novel framework is proposed for the weighted Meta graph-based Classification of Heterogeneous Information Networks (MCHIN) to address these challenges. The proposed framework has several appealing properties. In contrast to other proposed approaches, MCHIN can fully compute the weights of different meta graphs and mine the latent structural features of different nodes by using these weighted meta graphs. Moreover, MCHIN significantly enlarges the training sets by introducing the concept of Extension Meta Graphs in HINs. The extension meta graphs are used to augment the semantic relationship among the source objects. Finally, based on the ranking distribution of objects, MCHIN groups the objects into pre-specified classes. We verify the performance of MCHIN on three real-world datasets. As is shown and discussed in the results section, the proposed framework can effectively outperform the baselines algorithms.

**Keywords:** heterogeneous information networks; classification; meta graph; meta path

## 1. Introduction

In the real world, there exist lots of various entities, and together with their inter-relationships, they can be represented as information networks [1], such as Bibliographic Information Networks [2], Wikipedia [3], and Facebook. Existing studies [4–9] mainly focus on representing and analyzing such systems using homogeneous information networks, which composed of one type of nodes and edges. For example, Figure 1a illustrates a co-author homogeneous network consisting of "authors" as nodes and "publishing" relationships as links. However, the real-world systems are usually complex, mixed-type, and heterogeneous which can be more realistically represented by heterogeneous information networks. For example, there are different kinds of objects such as accounts, blogs, and friends, and different types of links such as publishing, forwarding and commenting between different pairs of objects in social networks. As an example, Figure 1b shows a bibliographic information network composed of multi-typed objects. There are authors (A), conferences (C), and papers (P) as nodes, and different relationships such as "publishing" and "writing" between different pairs of nodes.

One of the widely studied problems in Heterogeneous Information Networks (HINs) is classification considered as a semi-supervised learning approach, in which the labels of objects are predicted based on the structural properties or relationships between nodes. The classification can be

applied in different setting of applications of of HINs such as similarity search [1] and recommendation systems [10,11].

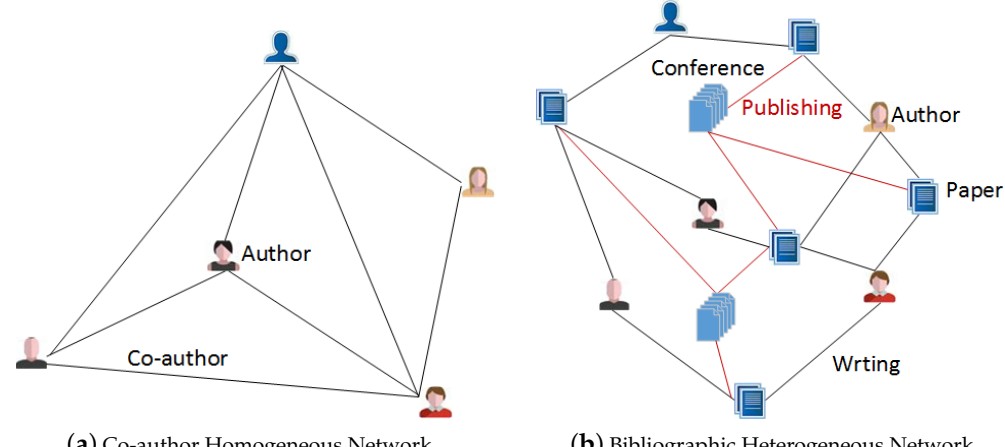

<div align="center">(<b>a</b>) Co-author Homogeneous Network      (<b>b</b>) Bibliographic Heterogeneous Network</div>

**Figure 1.** Co-authorship homogeneous network and bibliographic heterogeneous network. In the co-authorship homogeneous network (**a**), there is only one type of objects: authors, and single relationship: co-authorship; Bibliographic heterogeneous network (**b**), on the other hand, contains three types of objects: authors, papers and conferences. The red lines represent the publishing relationship, and the black lines express the writing relationships between authors and papers.

One of the most popular concepts used to study HINs is meta path. The meta path can better preserve the relationship information between different types of entities in HINs in comparison to other broadly used concepts such as random walks. However, due to the inherent limitation of meta paths such as their length [1], they can only carry limited semantics. Another problem is that typical meta path generation approaches can produce many short paths or even isolated nodes which make processing and learning even less efficient [12]. To tackle the weaknesses of meta paths, in reference [13], the concept of meta graphs is proposed, which contain different meta paths together and are shown to be significantly more useful in various tasks. Different meta graphs contribute differently to specific classification problems. However, to the best of our knowledge, none of the existing methods has demonstrated different semantics among meta graphs. Here, we compute the weights of different meta graphs by a priori knowledge.

On one hand, using labeled or annotated data can remarkably improve the classification performance, and on the other hand, acquiring labeled data or manual annotation of the available data is difficult and expensive. In this paper, we extend the labeled objects based on extending meta graphs and iteratively enlarge the training dataset. Finally, we use MCHIN to calculate the relatedness of multi-typed objects and group the objects into pre-specified classes by using the ranking distribution of objects. The contributions of this study are summarized as follows:

- We identify more objects for the training sets by the Extension Meta Graphs in HINs, which can effectively utilize a priori knowledge and improve the performance of classification.
- We introduce MCHIN which incorporates extension meta graphs' semantics and relevance measurement on objects and assigns categorical labels to each object by a ranking distribution function.
- We apply our proposed model on three real-world datasets and conduct a comprehensive analysis of the proposed model to gain more insights. The results show the effectiveness of MCHIN in the classification task in comparison to other state-of-the-art methods.

This paper is an extension of our previous conference paper "CHIN: Classification with META-PATH in Heterogeneous Information Networks [9]". We have significantly improved our

previous work and presented new meta graphs based classification of heterogeneous information networks algorithms. The main differences of our previous paper are shown as follows.

- We present a new model based on meta graphs, and compute the weights of different meta graphs.
- We introduce a novel extension rule to extend HINs, which is the first time to be proposed in our manuscript.
- We propose a novel framework MCHIN, which integrates the extend meta graphs and the weighted meta graphs to update the information of objects. Moreover, we provide the description of MCHIN algorithm.
- The datasets are different in the two papers. What is more, we use three datasets in our paper.
- We add Deepwalk, HIN2Vec, and SDNE as baselines in our experiments. We also introduce detailed information about baselines. We add the weights of different meta graphs. Besides, we experiment and theoretically analyze the reason why meta graphs are more effective than meta paths.

## 2. Related Work

Numerous research studies have been conducted on the classification of networks. The heterogeneous information networks have two main advantages over their homogeneous counterparts—they make it possible to model different types of objects and relationships, and they preserve more complex topological structures with richer semantics information [14].

**Classification of homogeneous information networks** There has been substantial attentions given to the classification of network data or objects recently. Most of the previous studies focus on the networks' structure, and others use unlabeled objects for classification [15–17]. Collective classification [18] is one of the most popular classification methods in mining networks. These methods classify objects by their features and the structure of the network in the homogeneous information networks. Zhou et al. [19] proposed the Learning with Local and Global Consistency (LLGC) algorithm. They design a function integrating sufficiently smooth coefficients to learn the general classification from the labeled data. The weighted-vote relational neighbor classifier (wvRN) [15,20] is another widespread classification algorithm proposed for mining network data. The wvRN label objects by considering a link selection mechanism.

**Classification of heterogeneous information networks** One of the algorithms proposed for mining HINs is GNetMine proposed by Ji et al. [21]. GNetMine is a transductive classification model based on graph regularization. The GNetMine, designed for HINs, can only handle the data with a common topic. Wan et al. [16] consider the class-level meta paths to construct more accurate classifiers and obtain the train labels to improve the active learning in HINs. RankClass algorithm, proposed by Ji et al. [17], is a more effective classification method that utilizes both classification and ranking operation. The values of ranking are related with the classification in HINs. Deepwalk [22] is another popular embedding model. The input to the Deepwalk is a series of objects produced by random walks in a language model-SkipGram, and the output would be the representation of features' objects. However, Deepwalk can only apply the short random walks to obtain the semantic among the objects.

## 3. Problem Formalization

The related concepts and definitions are introduced as follow.

We define a HIN as a graph denoted by $G = (V, E, A, R)$, where $V$ and $E$ represent the set of nodes and edges of the graph respectively, $A$ and $R$ represent the set of types of nodes and edges, respectively. Definition 1 represents network's meta-level structure, as shown in Figures 2a and 3a.

**Definition 1.** *Network schema [2,23]. The network schema is defined as $T_G = (A,R)$. $T_G$ is a meta template for network $G = (V, E, A, R)$ with the object type mapping $\phi : V \rightarrow A$ and the link type mapping $\psi : E \rightarrow R$.*

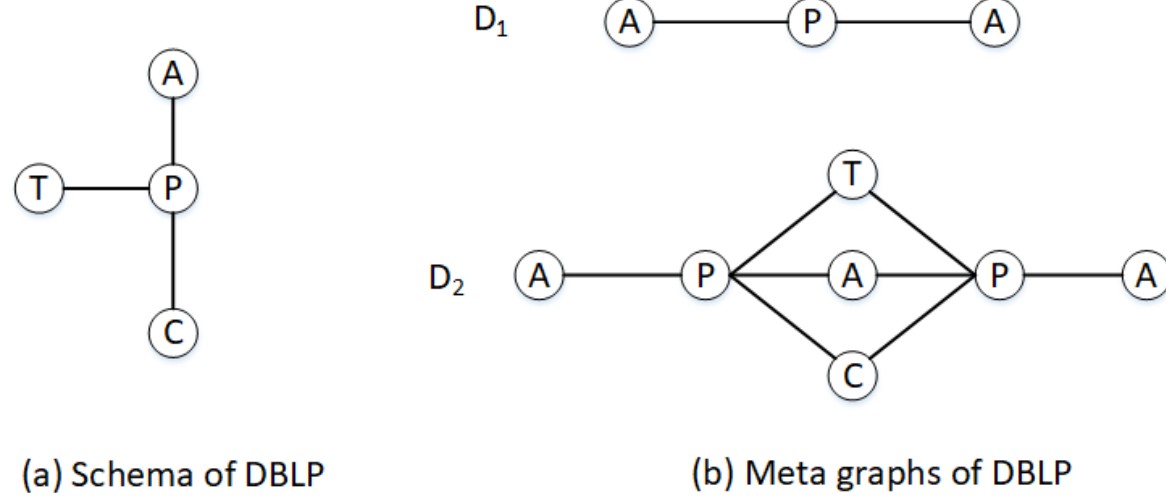

(a) Schema of DBLP

(b) Meta graphs of DBLP

**Figure 2.** Network Schema and Meta Graphs of DBLP (Digitall Bibliography & Library Project) Dataset.

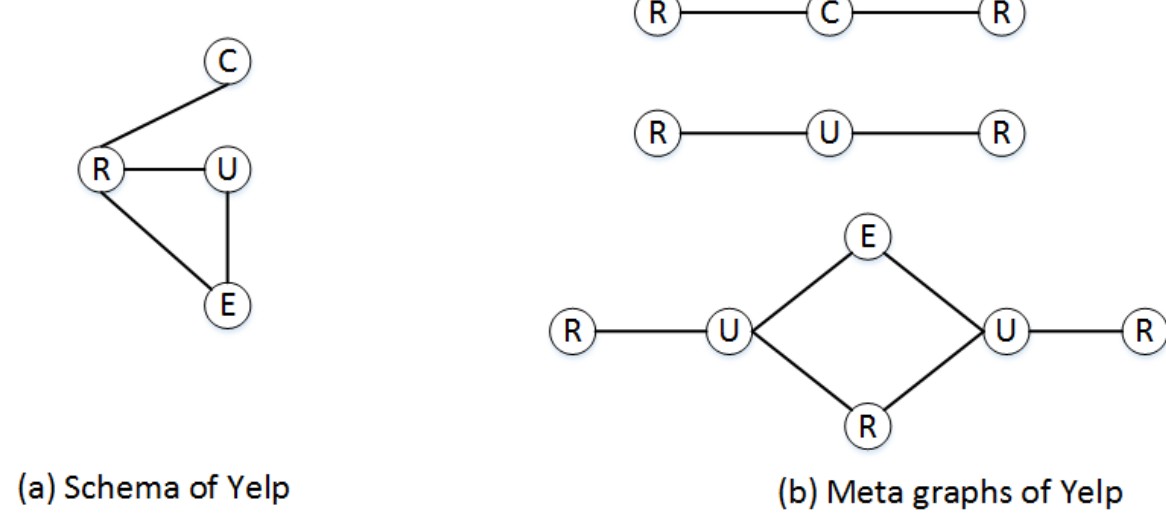

(a) Schema of Yelp

(b) Meta graphs of Yelp

**Figure 3.** Network Schema and Meta Graphs of Yelp Dataset.

In homogeneous networks, different paths connect two objects of the same type. However, in heterogeneous information networks, different paths can connect objects of different types. These connecting paths imply different semantics. Formally, we call these paths as meta paths.

**Definition 2.** *Meta path [1]. A meta path P is a composite relation $A_1 \xrightarrow{R_1} A_2 \xrightarrow{R_2} \ldots \xrightarrow{R_l} A_{l+1}$, which represents a composite relation $R = R_1 \circ R_2 \circ \cdots \circ R_l$, between object types $A_1, A_2, \ldots, A_{l+1}$, where $\circ$ denotes the composition operator on relations.*

Meta paths can express semantics between different objects. Obviously, different meta paths demonstrate different semantic meanings. In Figure 1b, meta path APA (Author-Paper-Author) shows that authors are co-writing a paper and the links between authors and papers denote co-author relationships.

**Definition 3.** *Classification in HINs. Given a class $C = \{c_1, c_2, \cdots, c_{|c|}\}$ and object set $V = \{v_1, v_2, \cdots, v_{|v|}\}$ in a HIN, where $|c|$ and $|v|$ denote the cardinality of these sets, respectively, then classification of HINs aims to map the object set V into the class set C through $f : V \to C$.*

**Definition 4.** *Meta graph [24]. A meta graph is denoted by $S = (A, L, n_t, n_s)$, where $A \subseteq V$ is a subset of nodes and $L \subseteq E$ is a subset of edges. The meta graph S is a directed acyclic graph (DAG), and $n_t, n_s$ are a single source and target node, respectively.*

Figure 2b shows the meta graphs of Figure 1b. In Figure 4a, we can consider three meta paths "$APA, APAPA, APVPA$" and a meta graph "$APA \bigcup AP(A; V)PA$". Tables 1 and 2 show respectively the connecting author neighbors of $A_1$ based on these meta paths and meta graph.

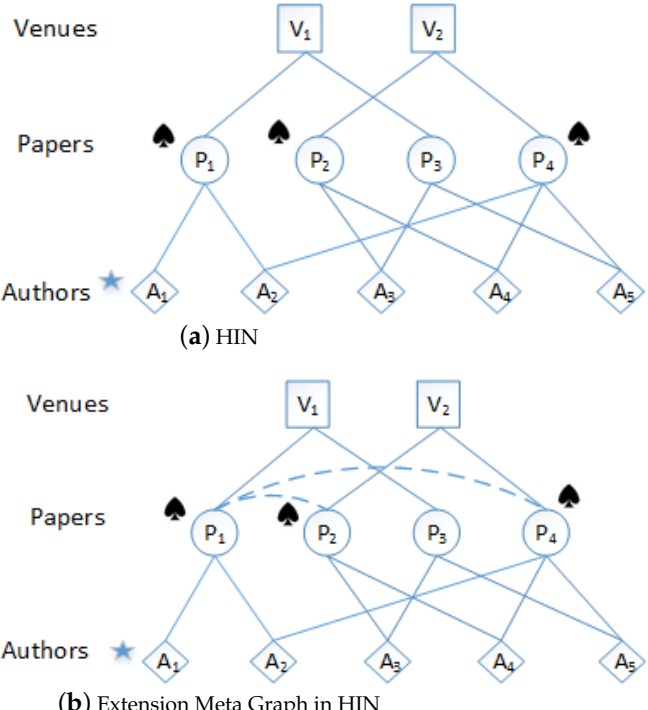

**(a)** HIN

**(b)** Extension Meta Graph in HIN

**Figure 4.** Extraction of Extension Meta Graph in HIN. (**a**) HIN; (**b**) Extension Meta Graph in HIN.

**Table 1.** The neighbors of $A_1$ based on three meta paths.

| Object | Meta Paths | Neighbors |
|:---:|:---:|:---:|
| | APA | $A_2$ |
| $A_1$ | APAPA | $A_4, A_5$ |
| | APVPA | $A_3, A_5$ |

**Table 2.** The neighbors of $A_1$ based on the meta graph.

| Object | Meta Graph | Neighbors |
|:---:|:---:|:---:|
| $A_1$ | $APA \bigcup AP(A; V)PA$ | $A_2, A_3, A_4, A_5$ |

**Definition 5.** *Meta graph matrix. A meta graph matrix M for a meta graph $S = (A_1 A_2 \cdots A_{l+1})$ is defined as $M_{1l+1} = W_{A_1 A_2} \times W_{A_2 A_3} \times \ldots \times W_{A_l A_{l+1}}$, where $W_{A_i A_j}$ is the adjacency matrix created for objects of type $A_i$ and $A_j$. $M[x_i, y_j]$ represents the number of graphs between objects $x_i \in A_l$ and $y_j \in A_{l+1}$ following meta graph S, and $M[x_i, y_j] = M[y_j, x_i]$.*

Obviously, meta graphs are composed of various meta paths in this paper. We take Figure 2b as the running example to better comprehend the computation of meta graph matrix. For meta graph $D_1$, it is computed as $M_{D_1} = W_{AP} \cdot W_{PA}$; the computing matrix of meta graph $D_2$ is computed as $M_{D_2} = W_{AP} \cdot \{(W_{PT} \cdot W_{TP}) \odot (W_{PA} \cdot W_{AP}) \odot (W_{PC} \cdot W_{CP})\} \cdot W_{PA}$, where "$\odot$" represents Hadamard product.

## 4. The MCHIN Algorithm

### 4.1. Extension of Heterogeneous Information Networks Based on Meta Graph

There are numerous influence relations between objects in heterogeneous information networks. Meta graphs involve a series of types of objects expressing binary relations defined on HINs to further enrich the relationships. However, we use the structure and semantics of objects in meta graphs to extract more information and improve the performance of the classification. Hence, we extend meta graphs to the extension meta graphs to capture more object correlations. The extension rules are as follows: (1) both objects are from the same type; (2) they are also given the same label. If the objects satisfy both of the rules, we link them with an edge. For convenience, the extension meta graphs can be expressed in the notation $\widetilde{A}$ for short.

To illustrate, consider the bibliographic heterogeneous network shown in Figure 4a. Here, four objects $P_1$, $P_2$, $P_4$, $A_1$ are already labeled (with three spades and a star). Without any extension, we can only relate $A_2$ to $A_1$ by APA (connecting $A_1$, $P_1$, and $A_2$). Now from the Extension Meta Graph $A\widetilde{P}A$, we can capture the relationship between $A_1$ and $A_3$, $A_4$ and $A_5$. According to meta graph $D_2$ in Figure 2, we can also link $P_1$ and $P_4$ by meta path $APAPA$ in $D_2$ (connecting $A_1$, $P_1$, $A_2$, $P_4$, and $A_5$) . In this way, We extract more information from HIN by the extension meta graphs.

### 4.2. Measuring the Similarity between Different Types of Objects

The relationships among different types of objects following the meta graphs are important in classification in HINs. Therefore, it is necessary to compute the similarity between the source and target objects following meta graphs in HINs. We introduce the details of similarity calculation as follows. Given a meta graph $S = (A_1 A_2 \cdots A_{l+1})$ with weight $\theta_s$, where $A_1$ and $A_{l+1}$ are different types of source object and target object, respectively, we want to calculate the relationship *Rel* between $a_{1i} \in A_1$ and $a_{(l+1)j} \in A_{l+1}$:

$$Rel(a_{1i}b_{(l+1)j} \mid S) = \frac{w(a_{1i}, a_{(l+1)j})\theta_s}{\theta_s \sum_j w(a_{1i}, a_{(l+1)j}) + \theta_s \sum_i w(a_{1i}, a_{(l+1)j})}, \tag{1}$$

where $w(a_{1i}, a_{(l+1)j})$ is the value of weighted matrix $M[a_{1i}, a_{(l+1)j}]$. In fact, $w(a_{1i}, a_{(l+1)j})$ is initially the number of meta graphs that connect $a_{1i}$ to $a_{(l+1)j}$.

### 4.3. Weight Learning of Meta Graph

To obtain the weight $\theta_p$ of each meta graph $s$, we construct the objective function $\mathcal{O}$ via a priori knowledge in HINs. The objective function $\mathcal{O}$ aims to maximize similarity with the same labels' objects, and minimize the similarity with the different labels' objects:

$$\mathcal{O} = \max_\theta \sum_{x_i, x_j \in \mathcal{L}} (sign(x_i, x_j) \sum_s Rel_s(x_i, x_j))^2 - \lambda \|\theta\|_2^2, \tag{2}$$

where $\lambda$ denotes the regularization parameter, $\| \cdot \|$ is defined as $\ell^2$ norm. $Rel_s$ is the similarity of objects following meta graph $s$. The definition of function $sign()$ is:

$$Sign(x_i, x_j) = \begin{cases} 1 & \text{if } x_i \text{ and } x_j \text{ share the same label} \\ -1 & \text{otherwise} \end{cases} \tag{3}$$

We calculate the partial derivatives $\theta_s$ of Equation (2), and obtain the solution of the loss function.

$$\theta_s = \frac{1}{2\lambda} \sum_{x_i, x_j \in \mathcal{L}} (Sign(x_i, x_j) \cdot Rel_s(x_i, x_j)) \tag{4}$$

### 4.4. The Framework of MCHIN

In a HIN, let $\mathbf{M}_{ij}$ be an $n_i \times n_j$ adjacent matrix of meta graph $s_{ij}$ with weight $\theta_r$. $M_{ij,pq}$ is the value of $p_{th}$ row and $q_{th}$ column in matrix $\mathbf{M}_{ij}$. And $M_{ij,pq}$ is also the weight of edges between objects $x_{ip}$ and $x_{jq}$. We consider undirected graphs taht satisfy $\mathbf{M}_{ij} = \mathbf{M}_{ji}^T$.

$$M_{ij,pq} = \begin{cases} 1 & \text{if } x_{ip} \text{ and } x_{jq} \text{ are adjacent} \\ 0 & \text{otherwise} \end{cases}$$

In a HIN, if objects have edges, then these objects have similar qualities such as the similarity rank scores. In the DBLP dataset, the higher of conferences' rank scores, the corresponding papers' rank scores would be higher. Therefore, we update each rank score of objects iteratively by the corresponding linked neighbors. The initial ranking distribution $P(x_{ip}|A_i,k)^0$ of objects is defined as a uniform distribution according to the labeled class k.

$$P(x_{ip}|A_i,k)^0 = \begin{cases} 0 & \text{otherwise} \\ \frac{1}{l_{ik}} & \text{if } x_{ip} \text{ belongs to } k\text{-th class,} \end{cases}$$

where $l_{ik}$ denotes the number of $A_i$ objects with $k$-th class.

Similar to References [21] and [25], linked objects are inclined to have similar labels according to the consistency assumption. In order to retain the consistency of pre-assigned labels, we add the priori knowledge to the objective function *obj*:

$$
\begin{aligned}
obj(P(x_{ip}|A_i,k)) &\doteq \lambda_{ij} \sum_{p=1}^{n_i} \sum_{q=1}^{n_j} \theta_r M_{ij,pq}(P(x_{ip}|A_i,k) - P(x_{jq}|A_j,k))^2 \\
&\quad + \alpha_i \sum_{p=1}^{n_i} (P(x_{ip}|A_i,k) - P(x_{ip}|A_i,k)^0)^2.
\end{aligned}
\tag{5}
$$

The first part of the function is the sum of the ranking distribution of objects with their neighbors, and the second part guarantees the consistency with the initial labels. When we minimize *obj*, it converges to the closed solution according to [16].

$$P(x_{ip}|A_i,k)^t \doteq \frac{\lambda_{ij}\theta_r M_{ij,pq}P(x_{jq}|A_i,k)^{t-1} + \alpha_i P(x_{ip}|A_i,k)^0}{\lambda_{ij} + \alpha_i}. \tag{6}$$

We iteratively normalize the $P(x_{ip}|A_i,k)^t$ as follows: $\sum_{p=1}^{n_i} P(x_{ip}|A_i,k)^t = 1, \forall i = 1,\ldots,m, k = 1,\ldots,K$.

The description of MCHIN's Algorithm 1 is as follows.

---

**Algorithm 1:** The algorithm of MCHIN

---

    **Input:** HIN $G = (V, E, A, R)$, meta graph $s$, meta graph matrix $M$

    **Output:** Weights of meta graphs $\theta_s$, the probability of each object belong to each class $k$

    begin Extend the HIN following meta graph $s$ based on extension rules, obtain extend meta
      graphs $\widetilde{s}$;

    Initialize the objects' ranking distribution within each class $k$;

    **while** *(not converge)* **do**

        Compute similarity of source objects and target objects following the extend meta graphs $\widetilde{s}$,
          obtain $Rel_{\widetilde{s}}$ with Equation (1);

        Compute the weight $\theta_{\widetilde{s}}$ according to Equation (4);

        Update the the ranking distribution within each class $k$ with Equation (6);

    **end**

    **end**;

    Compute each object's posterior probability ;

    **final** ;

    **return** Weights of meta graphs $\theta_s$, the objects labeled $k$ probability;

---

## 5. Experiments

To verify our model MCHIN in HINs, we chose three datasets—the DBLP "four-area" dataset (https://dblp.uni-trier.de) extracted from the bibliography database, Yelp (https://www.yelp.com/dataset/challenge) extracted from businesses website and IMDB dataset from the IMDB movie website (https://www.imdb.com/).

### 5.1. Description of Datasets

#### 5.1.1. DBLP Dataset

The DBLP dataset is a real-world computer science bibliography dataset. We model the DBLP dataset as a HIN in this experiment. There are four types of objects in DBLP, Author (A), Conference (C), Paper (P) and Keywords (K). Among different objects, there are various relationships, "writing" relationships between authors and papers, "publishing" relationships between conferences and papers, and "containing" relationships between keywords and papers. DBLP dataset contains four main conferences—Information Retrieval, Database, Artificial Intelligence, and Data Mining. The four conferences can be considered as the a priori knowledge. DBLP dataset involves 14,376 papers, 20 conferences, 14,475 authors, and 8920 keywords. There are 170,794 links in DBLP. The ground truth is 4057 authors and all 20 conferences. We choose the set $\mathcal{S} = (APA, AP\{A; C; T\}PA)$ as the meta graphs in our experiments. For the classification, the authors are labeled.

#### 5.1.2. YELP Dataset

Yelp is a website in which the users can find and evaluate different businesses. We extract a subnetwork that contains 4 types of objects—5000 restaurants (R), 257,953 Users (U), 10 Categories (C) ("American, Mexican, Italian, Chinese, Japanese, Thai, Indian, Canadian, Middle Eastern and Greek"), Reviews (E). Edges exist between restaurants and categories by the relation of "belong to", and categories are also the labels of businesses. The users are linked via "friendship" relationship. The classification task is to classify the restaurants by categories. In Figure 3, the meta graphs set is $\mathcal{S} = (RCR, RUR, RU\{E; R\}UR)$.

### 5.1.3. IMDB Dataset

We obtain the IMDB dataset from the movie website. IMDB movie network contains movies (M), actors (A), and directors (D). These three different types of objects form two main types of links: actors perform in movies (A-M) and directors direct movies (D-M). The movies genre is a priori knowledge. We divide the movies into three types of movies: drama, comedy, and actions in total. According to the semantic relationships between objects, we choose two proper meta graphs: $\mathcal{S} = (M\{A; D\}M, MA\{D; M\}AM)$ in IMDB dataset in our experiment.

### *5.2. Baselines*

To evaluate the effectiveness of our model MCHIN, we compare our model with five algorithms as follows:

- Deepwalk [22] adopts uniform random walks to represent network embedding for homogeneous networks. It conducted the SkipGram method to express nodes' features. Deepwalk can be applied to homogeneous information networks.
- HIN2Vec [26] is applied to HINs by representing the meta paths. HIN2Vec designs a neural network model and utilizes meta paths to perform classification task.
- SDNE [27] is a deep neural network based on a non-linear model. It joints the first-order and second-order proximity to present networks' structure.
- MCHIN-ori is MCHIN that does not contain extension meta graphs. The original MCHIN (MCHIN-ori) is used to evaluate the effectiveness of the extension meta graphs.

The introduction of LLGC, wvRN and RanClass can be found in related work. LLGC, Deepwalk and wvRN are all popular and widely used baseline methods. But all of the three models can only be applied to homogeneous information networks. In order to utilize these methods, we use different meta paths to transform HINs to homogeneous information networks.

### *5.3. Results and Analysis*

### 5.3.1. Classification of Nodes

In our experiments, we use classical method-accuracy to verify the effectiveness of MCHIN. We perform experiments on three real-world datasets—DBLP, Yelp, and IMDb. For more convincing results, we randomly select 3%, 5%, 7%, 10% labeled objects as the priori knowledge and classify the rest of the data in the DBLP dataset. Because Yelp dataset is very sparser than DBLP, we randomly select the proportion of labeled objects as $x$%(where $x = 10, 30, 50$ and $70$). In the IMDB dataset, we choose the ratio of labeled objects $x$%(where $x = 10, 30, 50$ and $70$) similar to the Yelp dataset.

In our work, $\lambda_{ij}$ indicates the selection of the important types of links during the ranking process. As discussed in Reference [17], we set $\alpha_i = 0.1, \lambda_{ij} = 0.2$. They are good enough to verify the validity of MCHIN.

The results are shown in Tables 3–5, and the highest performance is in bold. In DBLP, clearly, when the proportion of labeled nodes are only 3%, the accuracy of MCHIN can be up to 87.6%. That outperforms Deepwalk more than 18%, and better than the best baselines models by 7%. By increasing the proportion of labeled objects, the performance of the proposed MCHIN has been far ahead than other comparison methods. The accuracy of MCHIN can achieve 90.5% by labeling 10% of the objects. In fact, our proposed model MCHIN is significantly more effective than MCHIN-ori especially when there is a lower number of labels, as MCHIN uses extension meta graphs.

**Table 3.** Classification of nodes on DBLP dataset. The highest performance is in bold.

| Dataset | % Labeled-Nodes | 3% | 5% | 7% | 10% |
|---------|-----------------|------|------|------|------|
| DBLP | LLGC | 0.798 | 0.816 | 0.835 | 0.849 |
| | wvRN | 0.754 | 0.76 | 0.772 | 0.783 |
| | RankClass | 0.744 | 0.816 | 0.828 | 0.846 |
| | Deepwalk | 0.694 | 0.719 | 0.756 | 0.831 |
| | HIN2Vec | 0.786 | 0.817 | 0.845 | 0.863 |
| | SDNE | 0.802 | 0.824 | 0.851 | 0.872 |
| | MCHIN-ori | 0.862 | 0.869 | 0.887 | 0.902 |
| | MCHIN | **0.876** | **0.881** | **0.898** | **0.905** |

**Table 4.** Classification of nodes on Yelp dataset. The highest performance is in bold.

| Dataset | % Labeled-Nodes | 30% | 50% | 70% | 90% |
|---------|-----------------|------|------|------|------|
| Yelp | LLGC | 0.312 | 0.35 | 0.36 | 0.379 |
| | wvRN | 0.296 | 0.354 | 0.359 | 0.365 |
| | RankClass | 0.324 | 0.36 | 0.362 | 0.383 |
| | Deepwalk | 0.349 | 0.368 | 0.367 | 0.389 |
| | HIN2Vec | 0.360 | 0.382 | 0.396 | 0.411 |
| | SDNE | 0.374 | 0.389 | 0.396 | 0.415 |
| | MCHIN-ori | 0.363 | 0.403 | 0.409 | 0.416 |
| | MCHIN | **0.382** | **0.408** | **0.414** | **0.428** |

**Table 5.** Classification of nodes on IMDb dataset. The highest performance is in bold.

| Dataset | % Labeled-Nodes | 30% | 50% | 70% | 90% |
|---------|-----------------|------|------|------|------|
| IMDB | LLGC | 0.341 | 0.356 | 0.435 | 0.449 |
| | wvRN | 0.554 | 0.566 | 0.582 | 0.613 |
| | RankClass | 0.544 | 0.566 | 0.628 | 0.646 |
| | Deepwalk | 0.576 | 0.581 | 0.596 | 0.601 |
| | HIN2Vec | 0.616 | 0.630 | 0.641 | 0.687 |
| | SDNE | 0.354 | 0.368 | 0.376 | 0.451 |
| | MCHIN-ori | 0.662 | 0.669 | 0.727 | 0.732 |
| | MCHIN | **0.686** | **0.701** | **0.728** | **0.745** |

In Table 4, our method consistently outperforms the other baselines on the Yelp dataset as well. The results of LLGC, wvRN, RankClass and Deepwalk methods are quite similar to each other. The reasons are two-folds: (1) the four baseline methods are generally applied to homogeneous information networks. They lose the quality semantics and the useful complex information when they are applied to HINs. (2) None of the baseline approaches consider extra-label information through different types of models in HINs. When extending the meta graphs, MCHIN augments the prior label sets. This improves the performance 1% more than the MCHIN-ori method on average.

From Table 5, we can see that the effectiveness of MCHIN is higher than the best performance obtained by the baseline methods from 3% to 8% when the labeled objects change from 30% to 90%. The LLGC, wvRN, Deepwalk, and SDNE are generally used in homogeneous information networks, none of them can capture the rich semantic information in HINs. As the results of LLGC, wvRN, Deepwalk and SDNE methods show, they are significantly lower than others. Both of RanlClass and HIN2Vec algorithms are applied in the HINs, they can gain more semantic information in different relationships. Using meta paths in HIN2Vec effectively improve accuracy. However, the performance of MCHIN-ori is still up to 8% more than HIN2Vec because of adding meta graphs when the labeled objects are 70%. Furthermore, when we extend the meta graphs, the accuracy increases by 3% when the labeled objects are 50%.

5.3.2. Comparison of Algorithms Using Single Meta Path on DBLP

Meta graphs contain various meta paths, which represent different semantics in HINs. In order to verify the impacts of meta graphs, it is essential to compare the meta element-meta paths in meta graphs. Due to the space limitation, we only report the performance of meta graphs on the DBLP dataset. In our paper, the performances of four meta paths: $APA$, $APAPA$, $APTPA$, $APCPA$, and the meta graphs $\mathcal{S} = (APA, AP\{A; C; T\}PA)$ are compared for the DBLP dataset. Figure 5a–e show the accuracy of authors based on the meta paths and meta graphs. We can see that our model MCHIN outperforms all the baselines based on all the meta paths. The performance of MCHIN is still stable under different ratios of training data. In all the methods, meta path APCPA performs the best among APA and APAPA, as it can capture more semantic information in HINs.

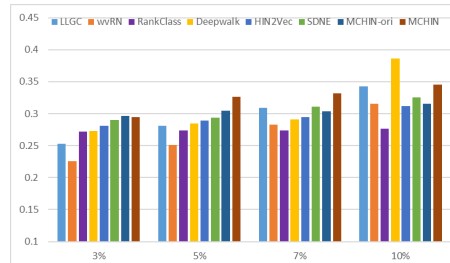

(**a**) Accuracy results based on the meta path APA

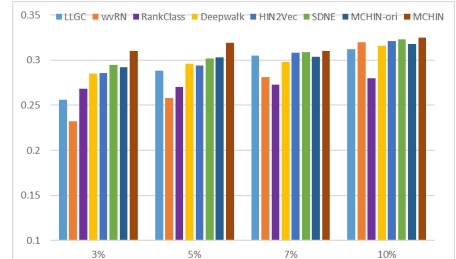

(**b**) Accuracy results based on the meta path APAPA

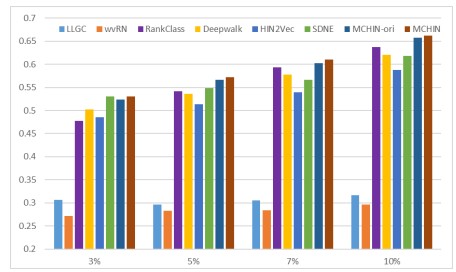

(**c**) Accuracy results based on the meta path APTPA

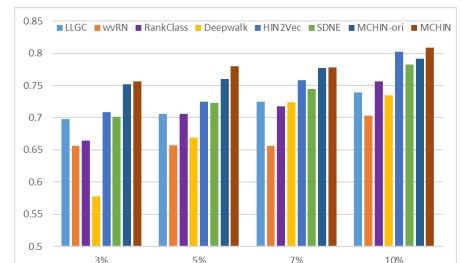

(**d**) Accuracy results based on the meta path APCPA

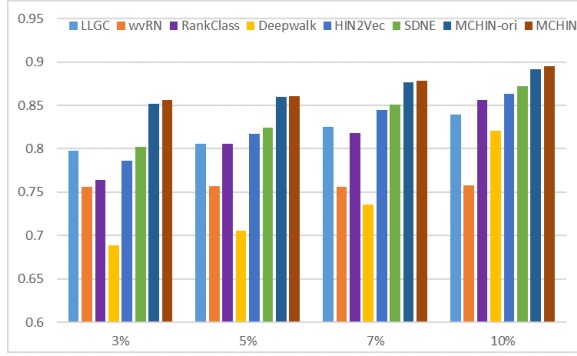

(**e**) Accuracy results based on the meta graph AP {A; C; T} PA

**Figure 5.** Results of accuracy of classification for author based on different meta paths and meta graph. (**a**) Accuracy results based on the meta path APA; (**b**) Accuracy results based on the meta path APAPA; (**c**) Accuracy results based on the meta path APTPA; (**d**) Accuracy results based on the meta path APCPA; (**e**) Accuracy results based on the meta graph AP {A; C; T} PA.

We further compare the performance of MCHIN based on meta graphs $\mathcal{S} = (APA, AP\{A; C; T\}PA)$ with all the four meta-paths. From Figure 5e, we can see that the meta graph $APA, AP\{A; C; T\}PA$ performs the best among the four meta paths. The single meta graph

$AP\{A;C;T\}PA$ outperforms other meta paths at least 1%. It shows that meta graphs can capture more semantic information than meta paths.

### 5.3.3. Learning Weights of Meta Graphs In MCHIN

From our experiments and comparisons, we can see that different meta graphs express different semantics. To utilize rich semantics of meta graphs, it is possible to weight meta graphs differently and assign higher weights to meta graphs with higher impacts on accuracy.

Figure 6 shows the performance of MCHIN under meta paths and meta graphs on DBLP. The meta graphs' relative effectiveness is determined by MCHIN via its weight assignment mechanism. Table 6 denotes different weights of meta graphs computed by MCHIN. MCHIN has assigned the highest weight to the meta graphs $AP\{A;C;T\}PA$, as $AP\{A;C;T\}PA$ performs the best in the classification. This matches with the intuition. On the other hand, the weight of APA is around 0.01~0.15 due to its poor ability to capture the main features for the classification task. The results of meta graphs' weights on the Yelp dataset are also shown in Table 6. Because the labels of the links between restaurants are related to categories, our model assigns more weights 0.30~0.40 to $RCR$ than $RUR$. The meta graph $RU(E;R)UR$ achieves the highest weight among other meta graphs. In the IMDB dataset, $MA\{D;M\}AM$ has higher weight in average than $M\{A;T\}M$, as the $MA\{D;M\}AM$ expresses more semantic than $M\{A;T\}M$.

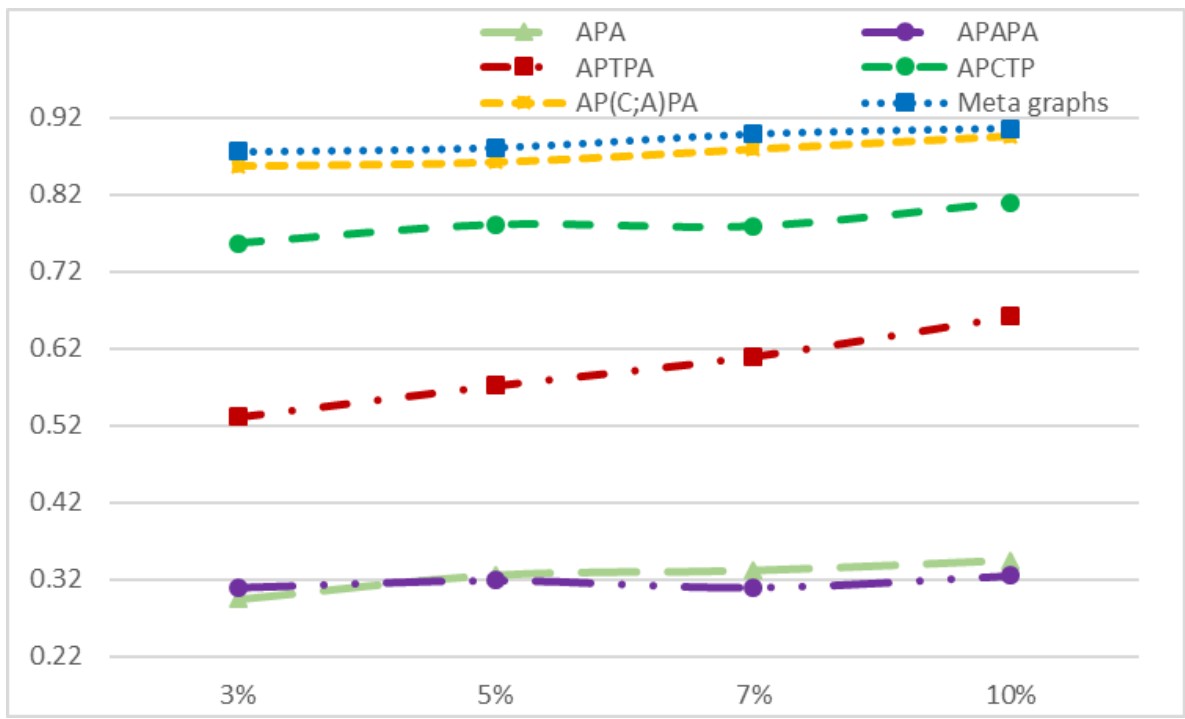

**Figure 6.** Accuracy comparison of MCHIN corresponding to different meta paths and meta graphs.

**Table 6.** Weights of meta graphs on DBLP, Yelp and IMDb dataset.

| Dataset | Meta Graphs | Weights |
|---|---|---|
| DBLP | APA | 0.01~0.15 |
|  | AP(A;C;T)PA | 0.85~0.90 |
| Yelp | RCR | 0.30~0.40 |
|  | RUR | 0.01~0.10 |
|  | RU(E;R)UR | 0.80~0.90 |
| IMDB | M(A;T)M | 0.45~0.55 |
|  | MA(D;M)AM | 0.70~0.90 |

## 6. Conclusions

In this paper, we studied the classification problem in HINs and proposed a new algorithm, MCHIN, that iteratively classifies objects in HINs. The priori knowledge is used to extend the original heterogeneous information networks, which can effectively capture the information hidden in semantic and structure. It consequently provides a richer training set and improves classification performance. The proposed framework explores the schema of the network to weight each meta graph, and integrate the weighted meta graphs for more effective classification. MCHIN also calculates the similarity of different objects to extract richer semantic information from the HINs. The performance results of experimental analysis based on different real datasets validate the superiority of MCHIN in comparison with other algorithms. In the future, we plan to apply our algorithm to other datasets. Another avenue for future research is to generalize the proposed method to semi-supervised problems, such as the multi-label classification problem in which an object in the network might have more than one label.

**Author Contributions:** Conceptualization, J.Z.; methodology, J.Z.; software, J.Z.; validation, J.Z., Z.J. and X.H.; formal analysis, T.L.; data curation, A.J.; writing—original draft preparation, J.Z.; writing—review and editing, T.L. and A.J.; supervision, Z.J. and X.H.; funding acquisition, X.H. All authors have read and agreed to the published version of the manuscript.

**Funding:** The authors would like to thank the anonymous reviewers for their valuable feedback. This work is supported by NSF IIS (No. 1815256), NSF IIS (No.1744661) and China Scholarship Council.

**Conflicts of Interest:** The authors declare no conflict of interest.

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
