# Peer review of "A Noval Weighted Meta Graph Method for Classification in Heterogeneous Information Networks"

_applsci, doi:10.3390/app10051603_

Round 1

Reviewer 1 Report

1. The classification problem is not properly described in 2.1. The author should give the formal description of the classification problem described in Section 1.

2. In 4.3, those methods could give different results according to how the labeled objects are chosen. The authors should repeat the experiments with different sets of selected labeled objects and show five number summaries of the results.

3. The authors should describe the aim of their design of the algorithm. The authors should show a motivating example in which their method works better than existing methods.

Author Response

Response

We would like to thank the reviewers for careful and thorough reading of our manuscript.  We sincerely thank the reviewers for constructive suggestions and valuable comments, which are of great help in revising the manuscript. We have carefully revised our manuscript with additional information and detailed explanation in order to further clarify previous confusions of our paper. Our detailed  responses and revisions are given below:

Response to reviewer 1:

The classification problem is not properly described in 2.1. The author should give the formal description of the classification problem described in Section 1.

Reply: As suggested by the reviewer, we give the formal description of classification problem in “Definition 3” on page “4”.

Definition 3. Classification in HINs. Given a classes set C = c1,c2,··· ,c|c|, and objects V = v1,v2,··· , v|v| in a HIN, where |c| and |c| denote the number of classes and objects, respectively. The classification of HINs aims to map the objects set V into the classes set C through f : V → C.

In 4.3, those methods could give different results according to how the labeled objects are chosen. The authors should repeat the experiments with different sets of selected labeled objects and show five number summaries of the results.

Reply: Thank you for the reviewer’s suggestion. Most papers that utilized DBLP dataset are all used the subset of DBLP website (https://dblp.uni-trier.de/). Our experiments use a specific subset of DBLP dataset, instead of different subsets. This is because most related works all use this particular subset, by focusing on this subset, we can compare our results with others. Here are a list of papers that use the DBLP subset of data.

“22  Perozzi, B.; Al-Rfou, R.; Skiena, S. Deepwalk: Online learning of social representations. Proceedings of the 20th ACM SIGKDD international conference on Knowledge discovery and data mining. ACM, 2014, pp.701–710.”

“Fan, Shaohua, Junxiong Zhu, Xiaotian Han, Chuan Shi, Linmei Hu, Biyu Ma, and Yongliang Li. "Metapath-guided Heterogeneous Graph Neural Network for Intent Recommendation." In Proceedings of the 25th ACM SIGKDD International Conference on Knowledge Discovery & Data Mining, pp. 2478-2486. 2019.”

“Dong, Yuxiao, Nitesh V. Chawla, and Ananthram Swami. "metapath2vec: Scalable representation learning for heterogeneous networks." In Proceedings of the 23rd ACM SIGKDD international conference on knowledge discovery and data mining, pp. 135-144. 2017.”

In our experiment, we choose different representative sets of labeled objects to verify our proposed model. There are lots of links between different types of objects in DBLP dataset, so we choose the ratios of objects are “3%, 5%, 7%, 10%”. As the other two datasets are sparser than DBLP dataset, so we select the ratios of objects are “30%, 50%, 70%, 90%”. The methods of choosing the labeled objects are the same with the above references.

The authors should describe the aim of their design of the algorithm. The authors should show a motivating example in which their method works better than existing methods.

Reply:

The purpose of our designing our experiments are as following:

The common tools for studying HINs are meta paths. However, our meta graphs are more effective than meta paths. To validate the effectiveness of meta paths, we compare meta paths with meta graphs in our experiment. What is more, different meta graphs has different effects in the classification task. It is important to compute the weights of different meta graphs by a priori knowledge in designing experiment. To verify the effectiveness of our proposed model, we experiment on different labeled objects. Obviously, it is better to verify the model on different training data than on the fixed training data. It can make our results more precision. The similar ratios of different labeled objects can be found in the following reference:

22 Perozzi, B.; Al-Rfou, R.; Skiena, S. Deepwalk: Online learning of social representations. Proceedings of the 20th ACM SIGKDD international conference on Knowledge discovery and data mining. ACM, 2014, pp.701–710.

Dong, Yuxiao, Nitesh V. Chawla, and Ananthram Swami. "metapath2vec: Scalable representation learning for heterogeneous networks." In Proceedings of the 23rd ACM SIGKDD international conference on knowledge discovery and data mining, pp. 135-144. 2017.

We agree and have made a motivating example in “Sec.2” on page “4”. Because we have already took examples about the necessary of extension of HINs, so we give an example of comparing meta paths with meta graphs in “Sec.2” on page “4”, which has been described in Table 1 and Table 2.

Reviewer 2 Report

The English language of the manuscript is poor. There are many sentences not correctly build in english language making it sometimes difficult for a reader to understand the ideas that the writers wanted to express. There are also many ortographic or grammar language mistakes. The manuscript should be carefully English-revised by a native or experienced writer.

Please change "priori knowledge" to "a priori knowledge" throughout all the manuscript.

Part of the authors recently published a very similar study (Ref. [9]). Please, explain explicitly in this new paper what's the contribution of that previous paper and what are the differences between this paper and Ref. [9].

Please, define "DBLP Dataset" when it first appears in the text (I think it' in Fig. 2 caption, page 3).

Line 133: Eq. (3.2 ?). Please correct Eq. number.

Alg. 1: Please correct Eq. numbers: (4.1), (4.4), (4.6).

Tables are not correctly numbered through all Section 4 in the text of the manuscript. Tables 2, 3, 4 and 5 are used instead of Tables 1, 2, 3 and 4.

Is Table 4 cited in the manuscript text?

Tables 1, 2, 3 and 4: Please indicate in table captions what are the figures indicated in the table (accuracy?). Also explain in the text of the manuscript how do you measure it.

Fig. 5: Please indicate color meaning in all figures.

Author Response

Response to reviewer 2:

We agree with the reviewer that sufficient care was not taken in the original manuscript vis-à-vis the English and interpretations. In the revised manuscript, the English and grammatical errors are corrected and the interpretations and flow have been significantly improved. As suggested by the reviewer, our manuscript has been carefully English-revised by a native writer.

Please change "priori knowledge" to "a priori knowledge" throughout all the manuscript.

Reply:  Thanks for pointing out such typos. The correction has been made in the revised manuscript.

Part of the authors recently published a very similar study (Ref. [9]). Please, explain explicitly in this new paper what's the contribution of that previous paper and what are the differences between this paper and Ref. [9].

Reply:  This paper is an extension of our previous conference paper “CHIN: Classification with META-PATH in Heterogeneous Information Networks[9]”. We have significantly improved our previous work and presented new meta graphs based classification of heterogeneous information networks algorithms. The main contributions of our previous paper are shown as follows.

We give the weights of each meta-paths, which can effectively promote the task of classification and address their limitations on the scarce labels. CHIN built the distribution over nodes ranking to determine an object’s optimal class membership. We conduct experiments to evaluate our algorithm on real datasets. The results show that our method is highly effective and achieve high classification precision.

The main differences of this paper with the previous version are:

We present a new model based on meta graphs, and compute the weights of different meta graphs. We introduce a novel extension rule to extend HINs, which is the first time to be proposed in our manuscript. We propose a novel framework MCHIN, which integrates the extend meta graphs and the weighted meta graphs to update the information of objects. Moreover, we provide the description of MCHIN algorithm. The datasets are different in the two papers. What is more, we use three datasets in our paper. We add Deepwalk, HIN2Vec, and SDNE as baselines in our experiments. We also introduce the detailed information of baselines. We add the weights of different meta graphs. Besides, we experiment and theoretically analysis the reason why meta graphs are more effective than meta paths.

Please, define "DBLP Dataset" when it first appears in the text (I think it' in Fig. 2 caption, page 3).

Reply:  Thanks for pointing out such typos. The correction has been made in the revised manuscript.

Line 133: Eq. (3.2 ?). Please correct Eq. number.

Reply:  We have carefully revised the numbering of equations according to the journal's format guidelines and corrected all the inconsistent numbering across the entire paper.

Alg. 1: Please correct Eq. numbers: (4.1), (4.4), (4.6).

Reply:  Thanks for pointing out such typos. The correction has been made in the revised manuscript.

Tables are not correctly numbered through all Section 4 in the text of the manuscript. Tables 2, 3, 4 and 5 are used instead of Tables 1, 2, 3 and 4.

Reply:  Thanks for pointing out such typos. The correction has been made in the revised manuscript.

Is Table 4 cited in the manuscript text?

Reply:  Yes, Table 4 (now Table 5) is cited at page 9. Every table has been cited in our revised manuscript.

Tables 1, 2, 3 and 4: Please indicate in table captions what are the figures indicated in the table (accuracy?). Also explain in the text of the manuscript how do you measure it.

Reply:  The correction has been made.

Fig. 5: Please indicate color meaning in all figures.

Reply:  As the graphic aesthetics in the Fig. 5 (a)-(d), we just indicate the color meaning in Fig. 5(e). There are two reasons

Each graphs of Fig. 5 (a)-(d) are half of Fig. 5(e). All graphs’ color meanings are the same in Fig. 5 (a)-(e).

    Thanks for your suggestions. We have indicated color meanings in Fig. 5 (a)-(e).

Round 2

Reviewer 1 Report

All the comments are well addressed.
I can now recommend accepting this paper.

I still have one concern.

As the authors write in their response to the other reviewer,
"This paper is an extension of our previous conference paper [9]”.
It is hard for readers to understand this. The authors should
make this clear.

Author Response

Response

We would like to thank the reviewer for careful and thorough reading of our manuscript.  We sincerely thank the reviewer for constructive suggestions and valuable comments again, which are of great help in revising the manuscript.

Our detailed  responses and revisions are given below:

As the authors write in their response to the other reviewer, "This paper is an extension of our previous conference paper [9]”. It is hard for readers to understand this. The authors should

make this clear.

Reply:  Thanks for pointing out this. We have significantly improved our previous work and presented new meta graphs based classification of heterogeneous information networks algorithms. The main differences of this paper with the previous version are:

1)  We present a new model based on meta graphs, and compute the weights of different meta graphs.

2)  We introduce a novel extension rule to extend HINs, which is the first time to be proposed in our manuscript.

3)  We propose a novel framework MCHIN, which integrates the extend meta graphs and the weighted meta graphs to update the information of objects. Moreover, we provide the description of MCHIN algorithm.

4)  The datasets are different in the two papers. What is more, we use three datasets in our paper.

5) We add Deepwalk, HIN2Vec, and SDNE as baselines in our experiments. We also introduce detailed information about baselines. We add the weights of different meta graphs. Besides, we experiment and theoretically analyze the reason why meta graphs are more effective than meta paths.

Thanks to the reviewer for reminding us about the ambiguity about our two papers. To make it clear, we add the main differences between the two papers in our manuscript (In Sec. 1 - “Introduction”).
